# Research Progress of Nattokinase in Reducing Blood Lipid

**DOI:** 10.3390/nu17111784

**Published:** 2025-05-24

**Authors:** Chuyang Wei, Ruitao Cai, Yingte Song, Xiaoyong Liu, Hui-Lian Xu

**Affiliations:** School of Biological Science and Technology, University of Jinan, Jinan 250024, China; wei_chaya@163.com (C.W.); crt17853319565@163.com (R.C.); 202321201498@stu.ujn.edu.cn (Y.S.)

**Keywords:** nattokinase, lowering blood lipid, mechanism of action, active ingredients

## Abstract

The purpose of this paper is to comprehensively review the research progress of nattokinase in lowering blood lipid, including its source, structure and physicochemical properties, mechanisms of functions, clinical research status, and safety considerations, so as to provide reference for further research on the application of nattokinase in the treatment of dyslipidemia. Natto is a traditional Japanese fermented food, which is made from soybeans fermented by Bacillus natto. During the fermentation process, natto will produce a variety of biologically active substances, including nattokinase. Nattokinase (NK) is a serine protease with stable enzyme activity and good freeze–thaw tolerance, which exerts lipid-lowering and anti-atherosclerotic effects by activating hormone-sensitive lipase (HSL), inhibiting hydroxymethylglutaryl monoacyl coenzyme A reductase (HMG-CoA reductase), and enhancing lipoprotein lipase (LPL) activity. Large-scale clinical trials have confirmed that nattokinase significantly improves the lipid profile and reduces the atherosclerotic plaque area and intima-media thickness with a favorable safety profile. Compared with traditional lipid-lowering drugs (e.g., statins and fibrates), nattokinase has a multifaceted lipid-lowering mechanism and lower risk of side effects, which makes it suitable for patients intolerant of traditional drugs; when combined with natural products such as statins, fibrates, red yeast, and lifestyle interventions, it can play a synergistic role and further reduce the risk of cardiovascular disease. There are various types of nattokinase preparations on the market, and consumers should choose regular products with high activity and purity, and pay attention to their safety and applicable population.

## 1. Introduction

Cardiovascular disease (CVD) is the leading cause of death worldwide. According to the World Health Organization, about 18 million people die of CVD every year, accounting for more than 30% of all deaths worldwide [1]. In China, the China Cardiovascular Disease Health and Disease Report 2021 shows that the prevalence of cardiovascular disease is on the rise, with two out of every five deaths due to cardiovascular disease [2]. Many bad life habits, such as staying up late, smoking, drinking, excessive obesity, stress, lack of exercise, and irrational diet, are gradually eroding the health of modern people and becoming important triggering factors for cardiovascular diseases. Bad living habits such as excessive drinking, irregular diet, smoking, and lack of exercise can lead to metabolic disorders and increase the body’s chronic diseases, such as hyperlipidemia, diabetes, and hypertension. Hyperlipidemia is caused by lipid metabolism disorders. Different from other chronic diseases, diabetes (type 1 diabetes, type 2 diabetes, gestational diabetes) is mainly caused by autoimmune cell destruction or failure and insulin resistance leading to high blood glucose levels [3]. Hypertension (primary hypertension, secondary hypertension), mainly by genetic factors, and kidney disease lead to arterial blood pressure continuing to rise [4]. For hyperlipidemia, bad living habits are an important reason for its occurrence.

Hyperlipidemia is divided into two categories, primary hyperlipidemia and secondary hyperlipidemia. Primary hyperlipidemia [5] includes familial hypercholesterolemia, familial mixed hyperlipidemia, and familial chylomicronemia. Familial hypercholesterolemia is due to LDLR, PCSK9, and APOB gene mutations that cause LDL clearance disorders, and ultimately LDL-C is significantly increased; familial mixed hyperlipidemia is a polygenic inheritance that leads to an increase in LDL and VLDL levels in the body, resulting in excessive TC and TG levels. Familial chylomicronemia is due to LPL, ApoC-II gene defects in the formation of the chylomicron accumulation of TG > 1000 mg/dL. There are many causes of secondary hyperlipidemia [6]. Basic metabolic diseases (such as diabetes, hypothyroidism, and nephritic syndrome) increase TG content in the body by affecting VLDL synthesis and lipoprotein synthesis. Unhealthy lifestyles (such as high saturated fat/trans fat diet, high sugar/alcohol intake, and lack of exercise) affect the normal metabolism of TG, resulting in excessive accumulation of TG. Nattokinase can also mobilize lipoprotein metabolism to promote TG hydrolysis by regulating lipid metabolism, effectively reducing TG accumulation in the body and achieving the effect of relieving hyperlipidemia.

In Japan, natto is an important part of the traditional diet; especially in the Kanto region (such as Ibaraki, Chiba), daily consumption is higher. As the active ingredient in natto, the intake of nattokinase is related to the following dietary habits and regional differences. The first is the regional difference. The consumption of natto in Guandong is significantly higher than that in Guanxi, which is related to the difference in the historical fermentation process (such as the traditional method of using straw wrapping to retain more nattokinase activity). Japanese children usually eat natto from early childhood, which is widely recommended as a supplementary food because of its high protein and easy digestion. Long-term intake may promote tolerance to nattokinase and potential cardiovascular benefits. According to the survey, about 70% of Japanese families eat natto more than 3 times a week, forming a natural and continuous pattern of nattokinase intake [7]. Regarding the availability of nattokinase, in non-traditional natto diet areas such as Europe and the United States, nattokinase is mainly obtained in the form of supplements, but there are the following problems, such as insufficient dose standardization, large differences in the activity (FU) of commercially available supplements, and some products not labeled with a clear titer. Cultural acceptance is low, and the special smell and taste of natto limit its popularity as food.

Nattokinase (NK) is a serine protease derived from natto and secreted by the bacterium Bacillus subtilis var. natto. Discovered in 1980 by the Japanese scholar Yoko Sumi, its potential efficacy in the field of cardiovascular health has attracted much attention [8]. As a dietary supplement, NK has a variety of physiological activities, among which the hypolipidemic effect is particularly prominent, which opens up a new way for the prevention and control of cardiovascular diseases. According to the study of Chen, H [9], the daily effective dose range of nattokinase is 2000–4000 FU (fibrinolytic unit), which is used to maintain cardiovascular health and take 2000 FU every day. With antithrombotic or thrombolytic needs, 4000 FU is taken every day (can be taken in batches), and short-term studies have shown that ≤10,000 FU/day has no serious side effects, but long-term high-dose data are limited. After analyzing the absorption mechanism of nattokinase, nattokinase is absorbed in the small intestine after oral administration. The peak blood concentration is about 2–4 h, and the half-life is about 8 h. It is recommended taking it in batches (such as 2000 FU in the morning and evening) to maintain the blood concentration [10]. Taking nattokinase on an empty stomach can absorb it faster but may stimulate the gastric mucosa. Taking nattokinase at night with slow blood flow can enhance the antithrombotic effect. Therefore, it is generally recommended taking it after dinners or before bedtime. In recent years, with the continuous deepening of the research in NK, many advances have been made in its mechanisms of functions, clinical effects, and application prospects. The aim of this review is to comprehensively sort out the lipid-lowering effects of NK.

The relevant research results of its use will provide a reference basis for further promoting its application in the prevention and treatment of cardiovascular diseases.

## 2. Characterization of NK

### 2.1. Chemical Structure and Physicochemical Properties

Nattokinase is a serine protease produced by Bacillus subtilis natto. The absorption pathway of nattokinase has not been fully elucidated, but there are relevant research clues. As a protein substance, it may be absorbed through the transcellular transport or paracellular pathway of the small intestinal mucosa. Some studies have pointed out that its structural stability is strong, and it may resist the degradation of gastrointestinal proteases in a complete molecular form to achieve absorption. After absorption, it enters the liver through the portal vein system, enters the systemic circulation after initial metabolism, and distributes to target tissues such as blood vessels. In terms of signal cascade reaction, nattokinase has multiple mechanisms of action: on the one hand, it can directly hydrolyze fibrin clots, activate plasminogen to convert into plasmin, and degrade fibrinogen and fibrin to dissolve thrombus; on the other hand, it can enhance the activity of the endogenous fibrinolytic system by activating tissue plasminogen activator (t-PA), while inhibiting plasminogen activator inhibitor-1 (PAI-1) to reduce fibrinolysis inhibition and improve the efficiency of thrombolysis through a dual mechanism. In addition, it may also improve vascular endothelial function by inhibiting inflammatory factors such as TNF-α and IL-6, and indirectly reduce the risk of thrombosis. The structure of NK is shown in Figure 1. NK belongs to the serine protease family, and its mature protein consists of 275 amino acid residues and is a disulfide bond-free single-chain polypeptidase with a molecular weight of about 27.7 kDa [11]. The three-dimensional structure was analyzed by X-ray crystal diffraction and nuclear magnetic resonance, and it is known that NK contains seven α-helices, nine β-folds, and two Ca^2+^-binding sites, as well as three tryptophan (Trp), twelve tyrosine (Tyr), and three phenylalanine (Phe) residues, which are structurally closely related to its enzymatic activity and stability [12]. On the level of molecular biology, DNA family shuffling is one of the ways to improve the fibrinolytic activity of nattokinase [13]. Site-directed mutagenesis is another means to enhance the activity and oxidative stability of NK [14,15]. During the NK purification process, the traditional protein expression and purification technology were changed and improved for better activity, but the results were less useful [16]. Beyond that, chemical modification has become an effective and popular approach to improve nattokinase activity and stability. NK was first well immobilized onto magnetic nanoparticles Fe_3_O_4_ in the presence of 1-[3-(dimethylamino)propyl]-3-ethylcarbodiimide (EDC), and it turned out that it had much higher thrombolytic activity, even higher than the pure NK [17]. Immobilization of nattokinase upon polyhydroxybutyrate (PHB) nanoparticles resulted in a 20% increase in the enzyme activity [18]. The adsorption capacity of nanosilver (AgNPs) on nattokinase enhanced the heat stability and anticoagulant effect of NK [19]. The effect of various metal ions, such as Ag(I),K(I), Na(I), Ba(II), Ca(II), Cd(II), Co(II), Cu(II), Hg(II),Mg(II), Mn(II), Ni(II), Zn(II), Fe(II), Fe(III), and Al(III), had been tested on enzyme activity with final concentrations from 1 to 10 mmol/L [20].

In terms of physicochemical properties, NK is in the form of a light-yellow solid, with good water solubility. Its isoelectric point (pI) is 8.6 ± 0.3, and its stability is better in the alkaline environment of pH 7–12. Especially the enzyme activity is the most stable at pH 7–9; when pH < 5, the enzyme protein is easy to be denatured and inactivated, but even under acidic conditions, NK still retains some residual activity [21]. Temperature has a significant effect on its activity, which is relatively stable in the solution below 45 °C, and after 60 °C, the enzyme protein is gradually denatured and rapidly loses its activity as the temperature rises. It is worth noting that NK well tolerates repeated freezing and thawing, with about 95% of the enzyme remaining active after five freeze–thaw cycles, which facilitates its production, storage, and transportation [22]. In addition, metal ions modulate NK activity: Hg^2+^ can cause complete inactivation of the enzyme; Mn^2+^, Zn^2+^, Co^2+^, and Cu^2+^ inhibit the enzyme activity to varying extents; while Ca^2+^ has an activating effect on enzyme activity at specific concentrations, exhibiting the property that its activity is finely regulated by external environmental factors [23]. Understanding the pH and temperature stability, isoelectric point, and freeze–thaw cycle tolerance of nattokinase is helpful to select appropriate ingredients and processing conditions in food formulation design. For example, select a suitable buffer system to maintain the pH value of the food within the stable range of nattokinase, or add a protective agent to improve its heat resistance and freeze–thaw cycle resistance. In the process of processing and storing food containing nattokinase, high temperature, acidic or alkaline environment, and multiple freeze–thaw cycles should be avoided to ensure the stability of nattokinase, for example, the use of low-temperature processing and storage technology, or through genetic engineering and other means, to improve the heat resistance of nattokinase and freeze–thaw cycle resistance. In addition, the activity of nattokinase is regularly detected during the production and storage of food containing nattokinase to ensure its stability in food. By understanding its physical and chemical properties, we can better formulate quality control standards and detection methods to ensure the functional properties and safety of food.

### 2.2. Sources and Production

Traditionally, NK is derived from Natto, a traditional Japanese fermented food with a long history of Natto production. Selected soybeans are pre-treated by washing, soaking, and autoclaving, and then inserted into Bacillus subtilis var. natto and fermented in a constant-temperature, high-humidity, partially aerobic environment at about 40 °C for 18–24 h, during which time Bacillus subtilis var. natto multiplies and secretes NK. This natural fermentation method endows natto with rich nutrients and physiological active substances, and makes NK work synergistically with other nutrients to ensure the safety and health efficacy of the product. However, the traditional fermentation has the problems of large batch-to-batch variations, low yield, unstable quality, and susceptibility to contamination by stray bacteria, which makes it difficult to meet the needs of modern industrialized large-scale production and precision medical treatment.

Along with the rapid development of modern biotechnology, microbial fermentation engineering and genetic engineering technologies have opened up a new path for the efficient production of NK. At the level of microbial fermentation, the yield and activity of NK can be greatly enhanced by optimizing the medium formulation, precisely regulating the types and proportions of carbon, nitrogen, inorganic salts, and growth factors, and combining with the automated control of fermentation temperature, pH, dissolved oxygen, stirring rate, and other conditions. For example, Pan et al. used the response surface optimization method to determine the optimal fermentation conditions, which can increase the yield of NK several times [24]. Meanwhile, genetic engineering technology realized the heterologous expression of NK. The NK gene is cloned into *E. coli*, Pichia yeast, and other expression hosts, with the help of gene editing means to optimize codons, enhance promoter activity, build fusion tags, etc., to achieve the efficient soluble expression of the protein, which not only improves the yield but also facilitates the subsequent isolation and purification [25], injecting a strong impetus for the industrialization of NK and promoting it from the traditional food field to the field of pharmaceuticals, health care, and other diversified application areas. NK can be utilized in a variety of applications, such as medicine and health products. NK production by natural fermentation mainly depends on the traditional natto fermentation process. NK is produced by natural fermentation of *Bacillus subtilis* var. *natto* under specific conditions. The advantage of this method is that the process is relatively simple and the cost is low, which is suitable for large-scale production in the food field. However, the NK activity produced during natural fermentation is relatively low, and it is usually difficult to meet the needs of high activity in the pharmaceutical field [26]. In addition, a variety of metabolites are produced during the natural fermentation process. The presence of these by-products will reduce the purity of NK, which may affect its application in the pharmaceutical field [27]. From the perspective of regulatory compliance, NK produced by natural fermentation is mainly used in food, and its regulatory requirements are relatively relaxed. However, if it is applied to the pharmaceutical field, it is necessary to meet stricter regulatory requirements such as GMP, which may increase additional costs and complexity. In contrast, recombinant production of NK is the use of genetic engineering technology to transform *Bacillus subtilis* var. *natto* to improve the yield and activity of NK. For example, by knocking out certain genes (such as the bdhA gene), unnecessary metabolites can be reduced, thereby increasing the purity of NK [28]. This recombinant production method can significantly improve the activity and yield of NK, making it more suitable for use in the pharmaceutical field. However, the cost of recombinant NK production is relatively high, and more resources need to be invested in genetic engineering, fermentation process optimization, and quality control. From the perspective of regulatory compliance, reorganized NK is mainly used in pharmaceuticals, so it is necessary to strictly comply with regulations such as the Good Manufacturing Practice (GMP) to ensure the safety, effectiveness, and quality control of products. This not only increases the supervision cost in the production process but also puts forward higher requirements for the quality management system of production enterprises.

## 3. Dangers of Dyslipidemia and Current Status

### 3.1. Health Problems Caused by Dyslipidemia

Dyslipidemia, especially the hyperlipidemic state, is a key risk factor for the development of atherosclerosis [29]. Excessive cholesterol, triglycerides, and low-density lipoprotein cholesterol (LDL-C) and other lipid components in the blood are easily deposited in the vascular endothelium, triggering an inflammatory response, prompting monocytes and smooth muscle cells to migrate and proliferate, and then forming atherosclerotic plaques. With the gradual increase and hardening of the plaque, the elasticity of the blood vessel wall decreases, the lumen narrows, and blood flow is impeded, leading to insufficient blood supply to tissues and organs of the body. This pathological change in the coronary artery manifests as coronary heart disease [30], coronary artery stenosis, or obstruction leads to myocardial ischemia and hypoxia, triggering angina pectoris, myocardial infarction, and other serious consequences; the cerebral blood vessels are prone to induce strokes, including ischemic and hemorrhagic strokes, which can lead to the patient’s limb hemiparesis, speech impediment, cognitive function decline, and even coma and death [31].

In addition, dyslipidemia is closely linked to a variety of metabolic disorders and health problems. It can cause fatty liver, excessive accumulation of fat in the liver, and interfere with the normal metabolism and detoxification function of the liver, causing long-term progression to liver fibrosis or cirrhosis; triglyceride levels are extremely elevated, which will increase the risk of the onset of acute pancreatitis; pancreatic tissue, due to inflammation infiltration damage, is life-threatening; at the same time, abnormalities of dyslipidemia are also related to insulin resistance, contributing to the elevation of blood glucose and increasing the incidence of diabetes, the formation of a vicious circle, further increasing the burden on the cardiovascular system, posing a serious threat to the overall health of the cardiovascular system. It further aggravates the burden on the cardiovascular system and poses a serious threat to overall health [32].

### 3.2. Epidemiologic Findings

A number of large-scale epidemiological surveys at home and abroad have revealed the severe prevalence of dyslipidemia. In China, according to the China Cardiovascular Health and Disease Report 2020, the overall prevalence of dyslipidemia among residents ≥35 years of age is as high as 34.7%, with certain differences in various regions, which are generally higher in the east coast than in the west inland, and higher in cities than in rural areas [33]. For example, the prevalence of dyslipidemia among adults in Beijing is about 40%, while certain less developed regions in the west are relatively lower, but also showing a rising trend year by year. From the age distribution point of view, the prevalence among middle-aged and elderly people is significantly higher than that among young people, and the proportion of dyslipidemia in elderly people over 60 years of age can be more than 50%, and males are higher than females. However, in recent years, with the change in lifestyle, the prevalence of dyslipidemia in young people is accelerating, and in children and adolescents, about 5–10% have borderline elevation of dyslipidemia, and obese children are particularly prominent.

Internationally, the problem of dyslipidemia in developed countries should not be underestimated. The National Health and Nutrition Examination Survey (NHANES) data show that the prevalence of dyslipidemia in adults is about 30–40%, and the prevalence rate in some European countries, such as Germany and the United Kingdom, is similar to that in the United States. In other Asian countries, Japan, and South Korea, with the economic development and westernization of the diet, the prevalence of dyslipidemia is also gradually rising, approaching the level of Europe and the United States. Dyslipidemia awareness, treatment, and control rates are generally low globally, and China’s awareness, treatment, and control rates are only about 16.1%, 7.8%, and 4.0%, respectively. Although developed countries have relatively higher rates, a large number of patients are not effectively managed, highlighting the urgency and necessity of strengthening the prevention and control of dyslipidemia [34].

## 4. Mechanism of Action of NK in Lipid Lowering

NK can directly act on the key link of lipid metabolism and regulate the blood lipid level. The mechanism of NK lipid-lowering is shown in Figure 2. Nattokinase can play a role in reducing blood lipids through the following four ways. First, it regulates lipid metabolism. Nattokinase activates hormone-sensitive lipase to promote the hydrolysis of triglycerides into free fatty acids and glycerol [35]. Second, nattokinase is involved in lipoprotein metabolism, enhancing lipoprotein lipase activity to accelerate the decomposition of triglycerides into fatty acids [36]. Third, nattokinase can inhibit the key enzymes of cholesterol synthesis to reduce endogenous cholesterol production and lower blood lipids [37]. Fourth, nattokinase has a good thrombolytic effect. Nattokinase can promote the production of plasminogen by endogenous plasminogen activator or activate plasminogen by the urokinase produced by the urokinase. Plasminogen is converted into fibrinolytic enzyme to dissolve the thrombus. At the same time, nattokinase can also directly hydrolyze cross-linked fibrin to dissolve the thrombus [38].

During lipolysis, it can activate the hormone-sensitive lipase (HSL) in adipocytes, prompting the hydrolysis of triglycerides into free fatty acids and glycerol, thus accelerating lipid mobilization in adipocytes and reducing fat accumulation in the body. Studies have shown that the addition of NK to the 3T3-L1 adipocyte culture system resulted in a significant increase in intracellular glycerol release and an elevated level of HSL phosphorylation, confirming its promotional effect on lipolysis [39]. In the regulation of cholesterol synthesis, NK inhibits the activity of hydroxymethylglutaryl monoacyl coenzyme A reductase (HMG-CoA reductase), a key enzyme for cholesterol synthesis in the liver [40]. By binding to the active center of this enzyme, it hinders substrate conversion and reduces endogenous cholesterol production. Animal experiments have revealed that the administration of NK intervention in a high-fat-diet-induced mouse model resulted in a decrease in hepatic HMG-CoA reductase expression and activity, and a subsequent decrease in plasma total cholesterol (TC) and low-density lipoprotein cholesterol (LDL-C) levels. For lipoprotein metabolism, NK enhances the activity of lipoprotein lipase (LPL), a key enzyme in triglyceride hydrolysis, which breaks down triglycerides in chylomicrons (CM) and very low-density lipoproteins (VLDL) into fatty acids for tissue uptake and utilization [41], and reduces plasma triglyceride levels. At the same time, NK promotes the hepatic uptake and metabolism of low-density lipoprotein (LDL), reduces its residence time in the blood, and lowers the risk of atherosclerosis; it can also stimulate hepatic synthesis of high-density lipoprotein (HDL), which can reverse cholesterol transport to the liver for metabolism, and exerts anti-atherosclerotic effects, and elevating the level of HDL can help to maintain the balance of blood lipids.

## 5. Evidence from Clinical Studies of Lipid Lowering by Nattokinase

### 5.1. Review of Early Small-Scale Clinical Trials

Early clinical trials on the lipid-lowering effects of NK were relatively small, with sample sizes ranging from a few dozen to more than a hundred cases, and study cycles usually ranging from a few weeks to a few months. Some studies focused on specific dyslipidemic populations, such as patients with primary hyperlipidemia or those with mild atherosclerosis.

In an 8-week trial enrolling 60 patients with primary hyperlipidemia, subjects were randomized into an experimental and a control group, with the experimental group receiving a daily preparation containing NK 2000 FU and the control group given a placebo. After a 56-day intervention, there were significant mean absolute changes in the total cholesterol (TC), low-density cholesterol (LDL-C), non-high-density cholesterol (non-HDL-C), and low-density cholesterol to high-density cholesterol ratio (LDL-C to HDL-C ratio), with values of −0.52 (95% CI: −0.51 to −0.54) mmol/L, −0.43 (95% CI: −0.45 to −0.41) mmol/L, −0.52 (95% CI: −0.52 to −0.52) mmol/L, and −0.29 (95% CI: −0.30 to −0.28) mmol/L, respectively, between the two groups. The results showed that serum total cholesterol (TC) levels in the experimental group were reduced by an average of about 10%, triglycerides (TGs) were reduced by about 12%, low-density lipoprotein cholesterol (LDL-C) was reduced by about 11%, and high-density lipoprotein cholesterol (HDL-C) was elevated by about 8% (*p* < 0.05, and the difference was statistically significant; in the control group, there was no significant change in the lipid indices) [42].

In another study, 30 patients with carotid intima-media thickening underwent a 3-month intervention, in which the experimental group was given NK (1500 FU/day) in combination with conventional lipid-lowering drugs, and the control group was given only conventional lipid-lowering drugs. At the end of the experiment, the reduction in carotid intima-media thickness (IMT) was more significant in the experimental group than in the control group; there was a significant reduction in CCAIMT and carotid plaque size in both groups compared with the baseline before treatment. The carotid plaque size and CCA-IMT reduced from (0.25 ± 0.12) cm^2^ to (0.16 ± 0.10) cm^2^ and from (1.13 ± 0.12) mm to (1.01 ± 0.11) mm, respectively. The reduction in the NK group was significantly profound (*p* < 0.01, 36.6% reduction in plaque size in NK group versus 11.5% change in stat).

### 5.2. Review of Early Large-Scale Clinical Trials

In recent years, with the in-depth research on NK, large-scale, multi-center clinical trials have been gradually launched, providing a more solid basis for its clinical application. A study published in *Frontiers in Cardiovascular Medicine* [43] is highly representative of the study, which retrospectively analyzed the data of 1062 participants between 2016 and 2020, who took a NK preparation product approved by China’s National Medicinal Products Administration (NMPA) and manufactured by Guangdong Shuangjun Biotechnology Co. (ShanTou, China) (NMPA); they took the approved NK preparation product manufactured by Guangdong Shuangjun Biotechnology Co. at a single dose of 3600 FU three times a day for a total of 10,800 FU/day for 12 months, with regular testing of blood lipids and carotid ultrasound.

The results of the study are impressive, resulting in atherosclerotic plaque patients with a 21.7% reduction in the thickness of the inner intima-media, with an effective rate of 77.7%, and a 36.0% reduction in the area of plaque, with an effective rate of 66.5%, indicating that NK can effectively inhibit the progression of atherosclerosis. After 12 months of daily NK consumption at a dose of 10,800 FU, a significant reduction in TG, TC, and LDL-C (*p* < 0.01) was evident compared to the values before treatment. Furthermore, NK also had the effect of increasing HDL-C (15.8% increase, *p* < 0.01). The levels of TC, TG, LDL-C, and HDL-C improved in 95.4, 85.2, 84.3, and 89.1% of the participants, respectively, after 12 months of NK use. NK administration for 12 months led to a decrease of 15.9, 15.3, and 18.1% in TC, TG, and LDL-C, respectively. In terms of lipid regulation, total cholesterol in hyperlipidemic patients was reduced by 15.9%, with an effective rate of 95.4%; tri-glycerides were reduced by 15.7%, with an effective rate of 85.2%; LDL cholesterol was reduced by 18.1%, with an effective rate of 84.3%; and HDL cholesterol was increased by 15.8%, with an effective rate of 89.1%, which comprehensively improved the lipid profile. No obvious adverse reactions related to the use of NK were found in the whole study, and there was no significant difference in the efficacy between male and female subjects, which fully confirmed the effectiveness and safety of this dose of NK in the prevention and treatment of cardiovascular diseases, and provided key evidence for the promotion of clinical application.

### 5.3. Analysis of Different Population Subgroups

An in-depth investigation of the lipid-lowering effect of NK in different population subgroups can help realize precision medicine. In terms of age, the elderly population (≥65 years old) is at high risk of dyslipidemia due to the slowing down of the body’s metabolism and the aging of blood vessels [44]. It was found that the lipid-lowering response to NK in elderly subjects was similar to that of the young and middle-aged population, but the relative benefit of the elderly group was more pronounced in terms of improving vascular endothelial function and decreasing atherosclerotic inflammatory indexes, which may be related to the fact that NK antioxidant and anti-inflammatory properties attenuate the chronic damage of the vasculature in the elderly.

Sex difference studies have shown that the prevalence of dyslipidemia is lower in women than in men before menopause due to the protective effect of estrogen on lipids, but cardiovascular risk rises after menopause when estrogen levels plummet [45]. Under NK intervention, postmenopausal women’s HDL-C elevation was slightly higher than men’s, while men’s LDL-C reduction was more significant, suggesting that there are subtle differences in the lipid metabolism response to NK between genders, or it may be related to hormone-regulated lipid synthesis and transporter pathways.

Body mass index (BMI) is an important influencing factor; people with BMI > 27.5 (overweight or obese) are often accompanied by insulin resistance and increased lipid metabolism disorders [46]. Clinical studies have shown that this group is more sensitive to the lipid-lowering effect of NK; taking the same dose of NK, overweight and obese people with TC, TG, and LDL-C decreased by 10–20% more than normal-weight people, because of their body adipose tissue secretion of excessive inflammatory factors interfering with lipid metabolism, the synergistic effect of anti-inflammatory and lipid pathway modulation properties by NK is more powerful.

Among the lifestyle factors, people who smoke and drink have impaired lipid metabolism and high levels of oxidative stress and inflammation [47]. An NK intervention test found that the degree of improvement of lipid indexes in this population is better than that of non-smokers and alcohol drinkers. On the one hand, NK directly reduces lipids, and on the other hand, it fights against vascular injuries caused by bad habits; after taking NK in regular exercisers, the aerobic metabolism of the body is strengthened, and fat consumption is promoted, which further amplifies the effect of NK in reducing lipids and improving vascular function, and provides precise guidance for the personalized application of NK in people with different lifestyles.

## 6. Comparison and Combination of NK and Other Lipid-Lowering Methods

### 6.1. Comparison with Conventional Lipid-Lowering Drugs

Conventional lipid-lowering drugs such as statins and fibrates are widely used in the field of lipid management, and each has its own advantages and disadvantages compared to NK.

Statins are the cornerstone of lipid-lowering therapy, potently lowering LDL-C levels by inhibiting HMG-CoA reductase, and large-scale clinical trials have confirmed their ability to significantly reduce the risk of cardiovascular events [48]. For example, atorvastatin and rosuvastatin can reduce LDL-C by 30–50%, which plays a key role in the primary and secondary prevention of atherosclerotic cardiovascular disease (ASCVD). However, some patients may have adverse reactions, such as elevated liver enzymes and muscle pain, after using the drug [49]; long-term use also has an increased risk of new-onset diabetes, and a certain percentage of patients are intolerant to statin, limiting its application.

Fibrates mainly target elevated triglycerides and low HDL-C anemia by activating peroxisome proliferator-activated receptor alpha (PPAR-α), accelerating VLDL catabolism, lowering triglyceride levels, and elevating HDL-C [50]. Like fenofibrate, they are indicated for hypertriglyceridemic patients with triglycerides > 5.65 mmol/L to prevent acute pancreatitis attacks. However, fibrates also carry potential risks of gastrointestinal distress, hepatic and renal impairment, rhabdomyolysis, and limited LDL-C reduction when used alone [51].

Compared with these drugs, NK has multiple mechanisms of lipid-lowering action. In addition to regulating the key enzymes of lipid synthesis and metabolism, it also has synergistic effects of anti-inflammatory, antioxidant, and blood rheology improvement. NK has outstanding advantages in terms of safety and tolerance. It is derived from natural fermentation, with rare adverse reactions and high safety for long-term use, making it suitable for people who are intolerant to traditional drugs. Clinical studies have shown that although the lipid-lowering effect of NK is not as fast and effective as that of high-intensity statin drugs, it has a considerable effect on patients with mild-to-moderate dyslipidemia, especially in regulating triglycerides and improving the HDL-C level, and can stabilize plaque and improve vascular endothelial function, which can provide multidimensional protection for lipid management, and it is a useful supplement to traditional lipid-lowering drugs.

### 6.2. Synergies of Joint Applications

The combination of NK with other lipid-lowering tools is expected to exert synergistic effects and enhance lipid regulation.

In terms of drug combination, in combination with statins, NK can enhance the lipid-lowering effect and make up for the lack of HDL-C elevation by statins, while its anti-inflammatory and antioxidant aspects alleviate the adverse effects of statin-induced liver injury and muscle damage. Animal experiments showed that the combination of atorvastatin and NK in rats on a high-fat diet resulted in a more significant improvement in blood lipids, reduction in hepatic histopathological damage, and decrease in serum inflammatory factors than in the statin-alone group. Combined with fibrates for mixed hyperlipidemia, it can lower triglycerides and strengthen LDL-C regulation, reduce the dose-dependent adverse effects of fibrates, and enhance the durability of lipid-lowering efficacy.

For natural product combination applications, red yeast rice is rich in natural statins, which complement the NK combination. NK dissolves thrombus and improves blood rheology, while red yeast inhibits cholesterol synthesis and synergistically lowers TC and LDL-C and raises HDL-C [52]. The study conducted by BY-HEALTH and others showed that the combined intervention of the two for 120 days resulted in the improvement of several lipid indices and the reduction of cardiovascular disease risk in dyslipidemic people, which provides an example of synergistic lipid lowering by functional foods.

In the synergy of lifestyle intervention, exercise is combined with NK, which promotes fat burning and improves cardiopulmonary function, and NK helps regulate lipid metabolism disorders after exercise, accelerates metabolite removal, reduces fatigue, and facilitates muscle recovery, thus enhancing lipid-lowering and health-promoting effects. Dietary control with NK involves a low-fat, high-fiber diet to limit exogenous lipid intake and NK regulation of endogenous lipid synthesis and metabolism, both internal and external, to strengthen the management of blood lipids for the comprehensive prevention and control of dyslipidemia, promoting the maintenance of cardiovascular health to open up a new way of thinking.

### 6.3. The Common Lipid-Lowering Food in Life

In today ‘s society, hyperlipidemia has become a common health problem, and regulating blood lipid levels through diet has gradually become a research hotspot. In recent years, many studies have shown that some common foods in life have significant lipid-lowering effects, providing people with a natural and effective intervention. Auricularia auricula is an underrated ‘super food‘. It is rich in water-soluble dietary fiber, which can effectively adsorb cholesterol and reduce its absorption in the body. Studies have shown that serum total cholesterol levels and low-density lipoprotein (LDL-C) levels were significantly reduced in patients with long-term consumption of black fungus, while high-density lipoprotein (HDL-C) levels were also increased [53]. As a marine vegetable, the brown algae polysaccharide component of kelp can regulate liver fat metabolism, reduce cholesterol synthesis, and promote its excretion, especially for reducing triglycerides [20]. Chinese chive has also attracted attention because of its significant hypolipidemic effect. Chinese chives contain a large amount of sulfur compounds and dietary fiber, which can activate the enzyme system in the body, promote lipid metabolism, adsorb cholesterol, and reduce its absorption in the intestine [54]. Clinical trials have shown that people who eat 2–3 cloves of garlic a day have significantly better blood lipid levels than the control group [55]. Polysaccharides and triterpenoids in edible and medicinal fungi, such as Lentinula edodes [56] and Cordyceps militaris [57], also regulate blood lipid metabolism and reduce blood lipid levels through a variety of mechanisms The lipid-lowering effect of these foods not only provides a new choice for patients with hyperlipidemia but also provides a scientific basis for the development of natural lipid-lowering foods. A partial list of lipid lowering foods is shown in Table 1. In the future, the potential of these natural foods in the prevention and control of hyperlipidemia will be further explored as more relevant studies are carried out.

## 7. Current Application Status and Safety Considerations of NK Products

### 7.1. Types of NK Preparations in the Market

There is a wide range of NK preparations in the market, covering capsules, tablets, softgels, granules, oral liquids, etc., to meet different consumer needs. Capsules are the most widely used dosage form because they can effectively protect NK from being destroyed by gastric acid and are convenient to take and easy to store; tablets are relatively low in production cost, which facilitates large-scale production and transportation; and softgels have a good sealing of the content, which can mask the special flavor of NK and enhance the taste.

From the brand perspective, there are many famous brands at home and abroad. Japan’s Kobayashi Pharmaceutical, Noguchi Medical Research Institute, and other brands have a long history, relying on Japan’s pioneering advantage in the field of NK research and development; product quality is stable and reliable, trusted by consumers, often using high-purity NK raw materials, with advanced production technology and strict control of product quality. Some U.S. brands are well known, such as Zelift, by virtue of its strong scientific research strength to push forward the innovation of the product research and development, such as the Zelift NK selection. For example, SALIFO Nattokinase selects highly active NK with a variety of synergistic ingredients to create a comprehensive cardiovascular care formula. Domestic brands such as BY-HEALTH and Tongrentang have also laid out the field of NK products and have gradually emerged in the market by taking advantage of their local resources and channels, and by combining with the health needs of Chinese people and launching products that are suitable for the body of Chinese people.

There are significant differences in the activity, purity, and formulation of NK between different products. In terms of activity, the activity of some imported high-end products can reach 40,000 FU/g or even higher, while the activity of some domestic ordinary products is only a few hundred FU/g, which is dozens of times different. Purity is related to the impurity residue: high purity products have less impurities, high bioavailability, and more significant efficacy; low purity products may be mixed with more fermentation by-products, affecting the safety and effectiveness. Formulations, in addition to a single NK preparation, include compound products, adding red, coenzyme Q10, deep-sea fish oil, and other ingredients to synergistically play the role of lipid-lowering and antioxidant, and improve vascular endothelial function. Product labeling of the applicable population is not the same. Part of the precision for the middle-aged and elderly, three-high population, cardiovascular disease recovery patients, and other specific groups is the detailed indication of the efficacy of the contraindications and recommendations for taking; a small number of products applicable to the description of the fuzzy are easy to mislead consumers. In view of this, consumers must choose to buy, through regular channels, well-known brands of products and read the product manuals according to their own conditions for reasonable choice.

### 7.2. Safety Assessment and Considerations

A large number of animal experiments and clinical trials have comprehensively evaluated the safety of NK. In animal experiments, mice, rats, and other experimental animals were given far more than the recommended dose of NK for human beings, and under continuous observation for several months, there was no obvious toxic reaction, no abnormal pathological changes in the histological examination of organs, and stable hematological and biochemical indexes, which indicated that NK has no acute or chronic toxicity hazards to the animal body under the normal dosage. Clinical trials have also confirmed its safety; in healthy people taking NK for a long time, the incidence of adverse reactions is very low, with occasional minor gastrointestinal discomfort, such as bloating, belching, etc., which after adjusting the way of taking or the dose of the symptoms can be relieved on their own. For patients with three highs and cardiovascular disease patients with combined drug therapy, NK does not increase the risk of additional adverse reactions and does not interfere with the effectiveness of conventional medications. Some studies also show that it can reduce the side effects of drugs, such as alleviating statin-induced liver injury.

## 8. Prospects for Future Research Directions

Nattokinase, as a natural bioactive substance derived from traditional fermented foods, has shown remarkable research results in the field of blood lipid reduction. Its multiple and unique mechanisms of action, including direct intervention in key aspects of lipid metabolism, potent anti-inflammatory and antioxidant properties, and optimization of blood rheology, synergistically regulate blood lipid levels in a multidimensional manner, opening up a new way of thinking in the prevention and treatment of cardiovascular diseases. After the early small-scale exploration to the recent large-scale, multi-center clinical studies, a large amount of data conclusively confirmed the significant effect of NK in lowering total cholesterol, triglycerides, and LDL cholesterol; raising HDL cholesterol; as well as inhibiting the progression of atherosclerosis, etc. And the safety of NK is good, with rare adverse effects, making it suitable for a wide range of people with high cardiovascular risk and those who need health improvements. Compared with traditional lipid-lowering drugs, NK has the advantages of being natural, mild, and multi-functional, which makes it a powerful supplement for lipid management, and when combined with other drugs, it can produce synergistic effects and reduce side effects.

On the application side, the rich and diverse NK preparations on the market provide consumers with multiple choices, with different dosage forms, brands, and formulas, but at the same time, it is also urgent for consumers to be prudent in their selection and to choose the appropriate products according to the formal channels and scientific instructions. In the future, with the flourishing of precision medicine, we should further deepen the research on the lipid-reducing mechanism and therapeutic efficacy of NK in different genetic backgrounds, lifestyles, disease subtypes, and other subdivided scenarios, so as to accurately locate the optimal beneficiary population and realize personalized and precise interventions. We should also innovate and optimize the formulation process with the help of cutting-edge technologies, such as nanotechnology and slow-release technology, so as to enhance the bioavailability and stability of NK, expand the routes of administration, and enhance the clinical efficacy of NK. We will also carry out more high-quality, long-cycle, and prospective clinical studies and combine the experimental data with a comprehensive assessment of the value of NK in the management of the whole course of cardiovascular diseases, so as to make NK a first-line weapon in the prevention of cardiovascular diseases and help the causes of global cardiovascular disease prevention and control, taking them to a new height.

In order to promote the in-depth development of natto lipid-lowering research, solve the problems existing in the current research, and improve the application effect of natto in the field of lipid-lowering, the future should break through from the following two aspects.

### 8.1. Strengthen Basic Research and Reveal the Mechanism of Action in Depth

Countries should increase investment in basic research on reducing blood lipids with natto and comprehensively use multi-omics technology to deeply analyze the mechanism of active components in natto. Through transcriptomics analysis, we comprehensively understood the changes in gene expression in the body after natto intervention, screened differentially expressed genes closely related to lipid metabolism, and clarified the regulatory role of these genes in the process of natto lipid-lowering. Using proteomics research, we identified differentially expressed proteins in biological samples before and after natto intervention, analyzed the functions and interaction networks of these proteins, and revealed the molecular targets and signaling pathways of natto for lowering blood lipids. Metabolomics research was carried out to detect the changes in metabolites in biological samples after natto intervention, to find biomarkers and metabolic pathways related to the lipid-lowering effect of natto, and to understand the mechanism of natto from the metabolic level. The research on the synergistic mechanism of each component in natto was strengthened. Through experimental design and data analysis, the interaction and synergistic effect between different components were clarified, which provided a theoretical basis for optimizing the formulation of natto products.

### 8.2. Conducting Large-Scale, Long-Term Clinical Trials

Researchers should organize large-scale, multi-center, long-term clinical trials to obtain more accurate and reliable clinical data on natto lipid lowering, as well as increase the sample size of clinical trials to ensure that the results can represent the characteristics and responses of different populations. By expanding the sample size, the impact of individual differences on the results of the study can be reduced, the statistical effectiveness of the study can be improved, and the lipid-lowering effect of natto in different populations can be more accurately evaluated. They should also prolong the cycle of clinical trials to observe the long-term effects of natto on blood lipid levels and whether long-term consumption of natto will produce potential adverse reactions. Long-term clinical trials can better simulate the practical application of natto and provide a more comprehensive assessment of the safety and efficacy of natto. In clinical trials, strict control of various confounding factors, such as the subjects’ lifestyle, diet, underlying diseases, etc., can ensure the accuracy and reliability of the results of the study. Standardized blood lipid detection methods and evaluation indicators were used to improve the comparability and repeatability of the research results. Strengthening the quality control and management of clinical trials can ensure that the research process meets ethical requirements and scientific norms.

## 9. Limitations of Research on Lipid-Lowering Effect of Nattokinase

The limitations of the research on the lipid-lowering effect of nattokinase are mainly reflected in the research design, mechanism of action, sample size, and clinical transformation. From the perspective of research design, most of the current studies are based on in vitro experiments or animal models. The number of human clinical studies, especially large-sample, long-term follow-up randomized controlled trials (RCTs), is limited, resulting in doubts about the reliability and universality of the conclusions. For example, the model induced by a high-fat diet, commonly used in animal experiments, is different from the pathophysiological mechanism of human hyperlipidemia, and the dose, dosage form (such as whether it is enteric-coated), and administration route of nattokinase may affect the experimental results. However, the relevant comparative studies are insufficient, and it is difficult to determine the optimal intervention plan.

### 9.1. Exploration of Mechanism of Action

The specific targets and signaling pathways of nattokinase in lowering blood lipid have not been fully elucidated. Most of the existing studies focus on its thrombolytic and anticoagulant effects, and there is a lack of systematic research on its direct effects on lipid metabolism (such as whether it regulates liver cholesterol synthesis, promotes fatty acid β-oxidation, or affects intestinal lipid absorption). Some studies have speculated that it may indirectly affect lipid metabolism by improving vascular endothelial function or anti-inflammatory effects, but the causal relationship has not been verified by mechanism experiments, resulting in a lack of solid theoretical basis.

### 9.2. Sample Size and Heterogeneity

Published human studies are mostly small samples (such as dozens of subjects), and the inclusion criteria are not uniform. There is a lack of stratified analysis of age, gender, underlying diseases (such as diabetes, hypertension), and medication history (such as statins), and it is difficult to exclude confounding factors. For example, some studies did not control lifestyle factors such as diet and exercise, resulting in the inability to accurately assess the independent role of nattokinase. In addition, the sources (such as fermentation products of different strains), purity, and dose of nattokinase in different studies vary greatly, and the results are not comparable, so it is difficult to form a consensus through meta-analysis.

### 9.3. Clinical Transformation Level

As a natural product, the quality control standards (such as enzyme activity units and impurity residues) of nattokinase have not been unified, and the effectiveness and safety of commercially available products are uneven, which may affect the repeatability of the research results. At the same time, the potential risks of long-term use of nattokinase (such as bleeding tendency and liver burden) lack data support, especially regarding research on the interaction with existing lipid-lowering drugs, which limits its clinical application.

## Figures and Tables

**Figure 1 nutrients-17-01784-f001:**
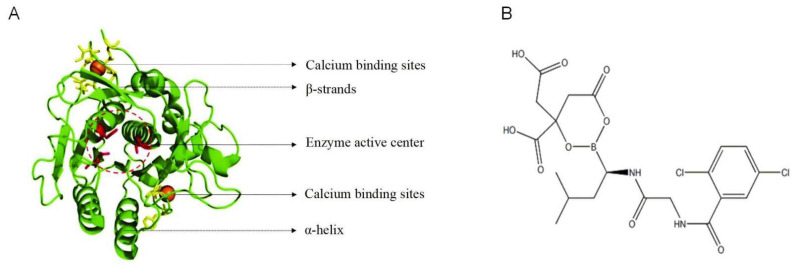
The structure of NK. (**A**) Spatial structure of Nattokinase; (**B**) Chemical structure of Nattokinase.

**Figure 2 nutrients-17-01784-f002:**
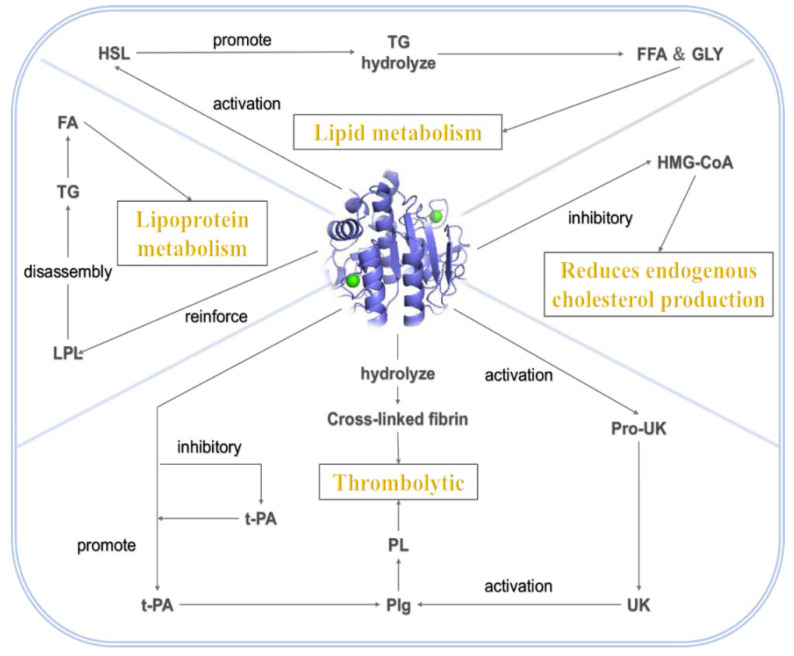
Nattokinase lipid-lowering mechanisms.

**Table 1 nutrients-17-01784-t001:** Some foods that can lower blood lipids.

Food Name	Bioactive Components	Mechanisms	Ref.
Natto	Nattokinase	Serum triglyceride, total cholesterol, and low-density lipoprotein cholesterol (LDL-C) levels were significantly reduced.	[58]
Mulberry	Ethyl acetate	Reduces serum triglyceride (TG), total cholesterol (TC), and low-density lipoprotein cholesterol (LDL-C) levels, thereby preventing atherosclerosis.	[59]
Citrus	Naringin	Increases anti-lipid peroxidation in liver, decreases SOD activity and oxygen free radical generation, and significantly reduces MDA content.	[60]
Sea Buckthorn Fruit	Quercetin	Promotes the conversion of cholesterol into bile acids, promotes cholesterol efflux, inhibits cholesterol de novo synthesis, and accelerates fatty acid oxidation.	[61]
Hawthorn	Hypericin and Ursolic acid	Can prevent the accumulation of fat in the liver, affect the PPARr-PPRE signaling system, and inhibit the synthesis of endogenous lipids.	[62]
Garlic	Garlicn	The levels of serum TC, TG, and LDL-C were significantly decreased, and the content of high-density lipoprotein cholesterol (HDL-C) was increased. The activities of serum lecithin cholesterol acyltransferase, myocardial lipoprotein lipase, and hepatic lipase were increased to varying degrees.	[63]
Onion	Onion alcohol extract	Increases the expression of LDLR protein and reduces the expression of HMG CR protein to play its lipid-lowering effect.	[64]
Capsicum	Capsaicin	Promotes the secretion of the nerve conduction substances acetylcholine and thyroid gland, and promotes body fat oxidation.	[65]
Tomato	Lycopene	Inhibits acyl-CoA.	[66]
Yam	Yam polysaccharide	Improves the activity of total superoxide dismutase (T-SOD), catalase, insulin sensitivity, and antioxidant activity.	[67]

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
