# Peer review of "Research Progress of Nattokinase in Reducing Blood Lipid"

_nutrients, 2025, doi:10.3390/nu17111784_

Round 1

Reviewer 1 Report

Comments and Suggestions for Authors

Nutriets-3644707: Research Progress of Nattokinase in Reducing Blood Lipid, authored by Chuyang Wei, et al.

General comments:

This paper is a review the research progress of nattokinase in lowering blood lipid, including its source, structure and physicochemical properties, mechanisms of functions, clinical research status and safety considerations for further research on the application of nattokinase in the treatment of dyslipidemia. The authors suggested that nattokinase effectively improves lipid profile and reduces atherosclerotic risks compared with traditional lipid-lowering drugs.  This manuscript includes interesting issues to the preventive medicine and nutritious field; however, the profound pathophysiological approach would be needed to demonstrate the effectiveness of nattokinases.

Specific comments:

  1. The authors focuses on the lipid-lowering effects of nattokinases; however, the metabolic disorders include not only hyper- or dys-lipidemia, but also diabetes, obesity, hypertension and chronic kidney diseases should be closely related each other.The authors need to clarify theses complexed conditions of many of the lifestyle-related diseases.
  2. The clinical types of hyperlipidemia should be classified to discuss the effectiveness of nattokinases.Also, the required daily contents, absorptive rates and the recommended durations for taking nattokinases should also be addressed.
  3. The absorptive pathways and the signaling cascades of nattokinases, and the metabolic process with half-life would be addressed.The ingredients for efficaciously taking nattokinases should be compared based on the literatures.
  4. Regarding the pathophysiology and the mechanical effects of nattokinases, the authors mentioned the molecular targets and the metabolic pathways related to the lipid-lowering effects of natto; however, no specific references were cited.
  5. Taking nattokinases should be dependent on the food culture and regional issues even in Japan.The authors need to discuss the availability of nattokinases and the food cultures for taking nattokinases from the childhood.

Author Response

Comments 1: [The authors focuses on the lipid-lowering effects of nattokinases; however, the metabolic disorders include not only hyper- or dys-lipidemia, but also diabetes, obesity, hypertension and chronic kidney diseases should be closely related each other. The authors need to clarify theses complexed conditions of many of the lifestyle-related diseases.]

Response 1: Thank you for pointing this out. We agree with this comment. Therefore, we have made the following change .[Bad living habits such as excessive drinking, irregular diet, smoking and lack of exercise can lead to metabolic disorders and increase the body's chronic diseases such as hyperlipidemia, diabetes, and hypertension. Hyperlipidemia is caused by lipid metabolism disorders. Different from other chronic diseases, diabetes (type 1 diabetes, type 2 diabetes, gestational diabetes) is mainly caused by autoimmune cell destruction or failure and insulin resistance leading to high blood glucose levels [1]. Hypertension (primary hypertension, secondary hypertension), mainly by genetic factors, kidney disease led to arterial blood pressure continues to rise [2]. For hyperlipidemia, bad living habits are an important reason for its occurrence.]

we have add the following in lines 34-43 “[updated text in the manuscript if necessary]”

Comments 2: [The clinical types of hyperlipidemia should be classified to discuss the effectiveness of nattokinases. Also, the required daily contents, absorptive rates and the recommended durations for taking nattokinases should also be addressed.]

Response 2: 

Thank you for pointing this out. We agree with this comment. Therefore, we have add the following, on page 3, lines 128-141.

Hyperlipidemia is divided into two categories, primary hyperlipidemia and secondary hyperlipidemia. Primary hyperlipidemia [3] includes familial hypercholesterolemia, familial mixed hyperlipidemia, and familial chylomicronemia. Familial hypercholesterolemia is due to LDLR, PCSK9, APOB gene mutations that cause LDL clearance disorders and ultimately LDL-C is significantly increased; familial mixed hyperlipidemia is a polygenic inheritance that leads to an increase in LDL and VLDL levels in the body, resulting in excessive TC and TG levels. Familial chylomicronemia due to LPL, ApoC-II gene defects in the formation of chylomicron accumulation of TG >1000 mg/dL. There are many causes of secondary hyperlipidemia [4]. Basic metabolic diseases (such as diabetes, hypothyroidism, nephritic syndrome) increase TG content in the body by affecting VLDL synthesis and lipoprotein synthesis. Unhealthy lifestyles (such as high saturated fat/trans fat diet, high sugar/alcohol intake, and lack of exercise) affect the normal metabolism of TG, resulting in excessive accumulation of TG. Nattokinase can also mobilize lipoprotein metabolism to promote TG hydrolysis by regulating lipid metabolism, effectively reduce TG accumulation in the body and achieve the effect of relieving hyperlipidemia.

we have add the following in lines 44-58

Comments 3: [The absorptive pathways and the signaling cascades of nattokinases, and the metabolic process with half-life would be addressed. The ingredients for efficaciously taking nattokinases should be compared based on the literatures.]

Response 3: 

Thank you for pointing this out. We agree with this comment. Nattokinase is a serine protease produced by Bacillus subtilis natto. The absorption pathway of nattokinase has not been fully elucidated but there are relevant research clues. As a protein substance, it may be absorbed through the transcellular transport or paracellular pathway of the small intestinal mucosa. Some studies have pointed out that its structural stability is strong, and it may resist the degradation of gastrointestinal proteases in a complete molecular form to achieve absorption. After absorption, it enters the liver through the portal vein system, enters the systemic circulation after initial metabolism, and distributes to target tissues such as blood vessels. In terms of signal cascade reaction, nattokinase has multiple mechanisms of action : on the one hand, it can directly hydrolyze fibrin clots, activate plasminogen to convert into plasmin, degrade fibrinogen and fibrin to dissolve thrombus ; on the other hand, it can enhance the activity of endogenous fibrinolytic system by activating tissue plasminogen activator ( t-PA ), while inhibiting plasminogen activator inhibitor-1 ( PAI-1 ) to reduce fibrinolysis inhibition, and improve the efficiency of thrombolysis through a dual mechanism. In addition, it may also improve vascular endothelial function by inhibiting inflammatory factors such as TNF-α and IL-6, and indirectly reduce the risk of thrombosis.

we have add the following in lines 101-116

According to the study of Chen, H [5], the daily effective dose range of nattokinase is 2,000-4,000 FU (fibrinolytic unit), which is used to maintain cardiovascular health and take 2,000 FU every day. With antithrombotic or thrombolytic needs, 4,000 FU is taken every day (can be taken in batches), and short-term studies have shown that ≤10,000 FU/day has no serious side effects, but long-term high-dose data is limited. After analyzing the absorption mechanism of nattokinase, nattokinase is absorbed in the small intestine after oral administration. The peak blood concentration is about 2-4 hours, and the half-life is about 8 hours. It is recommended taking it in batches (such as 2,000 FU in the morning and evening) to maintain the blood concentration [6]. Taking nattokinase on an empty stomach can absorb faster but may stimulate the gastric mucosa. Taking nattokinase at night with slow blood flow can enhance the antithrombotic effect. Therefore, it is generally recommended taking it after dinners or before bedtime.

we have add the following in lines 81-93

Comments 4: [Regarding the pathophysiology and the mechanical effects of nattokinases, the authors mentioned the molecular targets and the metabolic pathways related to the lipid-lowering effects of natto; however, no specific references were cited.]

Response 4: 

Thank you for pointing this out. We agree with this comment. Therefore, we have Add the following, on reference.

Comments 5: [Taking nattokinases should be dependent on the food culture and regional issues even in Japan. The authors need to discuss the availability of nattokinases and the food cultures for taking nattokinases from the childhood.]

Response 5: 

Thank you for pointing this out. We agree with this comment. Therefore, we have add the following, on lines 59-75.

In Japan, natto is an important part of the traditional diet, especially in the Kanto region (such as Ibaraki, Chiba) daily consumption is higher. As the active ingredient in natto, the intake of nattokinase is related to the following dietary habits and regional differences. The first is the regional difference. The consumption of natto in Guandong is significantly higher than that in Guanxi, which is related to the difference of historical fermentation process (such as the traditional method of using straw wrapping to retain more nattokinase activity). Japanese children usually eat natto from early childhood, which is widely recommended as a supplementary food because of its high protein and easy digestion. Long-term intake may promote tolerance to nattokinase and potential cardiovascular benefits. According to the survey, about 70 % of Japanese families eat natto more than 3 times a week, forming a natural and continuous pattern of nattokinase intake [7]. Regarding the availability of nattokinase, in non-traditional natto diet areas such as Europe and the United States, nattokinase is mainly obtained in the form of supplements, but there are the following problems, such as insufficient dose standardization, large differences in the activity (FU) of commercially available supplements, and some products are not labeled with a clear titer. Cultural acceptance is low, and the special smell and taste of natto limit its popularity as food.

Reviewer 2 Report

Comments and Suggestions for Authors

The manuscript addresses a relevant and timely topic regarding the lipid-lowering potential of nattokinase. It covers mechanisms, clinical findings, and practical applications in an informative manner. However, several major revisions are required.

First, the manuscript contains numerous grammatical errors, typographical issues, and awkward sentence constructions that hinder clarity. Professional language editing is recommended.

Second, figures and tables are poorly integrated; captions are minimal, and their significance is not clearly explained in the text.

Third, the references include outdated sources and inconsistent formatting—replacement with recent, peer-reviewed literature is advised.

Fourth, while the cited clinical results are promising, key statistical data such as p-values, effect sizes, and confidence intervals are missing.

Additionally, the limitations of the studies are not adequately discussed. Reorganizing the manuscript for logical flow and providing a more critical evaluation of the data will strengthen its scientific value. I recommend major revision before further consideration.

Comments on the Quality of English Language

The manuscript contains numerous grammatical errors, typographical issues, and awkward sentence constructions that hinder clarity. Professional language editing is recommended.

Author Response

Response 1: Thank you for pointing this out. We agree with this comment. Therefore, we have made the following change [updated text in the manuscript if necessary]”

Comments 2: [The figures and tables are poorly integrated; captions are minimal, and their significance is not clearly explained in the text.]

Response 2: 

Thank you for pointing this out. We agree with this comment. Therefore, we have add the following,, lines 285-296.

Comments 3: [The references include outdated sources and inconsistent formatting—replacement with recent, peer-reviewed literature is advised.

Response 3: 

Thank you for pointing this out. We agree with this comment.We have updated all references to ACS format. It can be seen in the manuscript.

Comments 4: [while the cited clinical results are promising, key statistical data such as p-values, effect sizes, and confidence intervals are missing. ]

Response 4: 

Thank you for pointing this out. We agree with this comment. Therefore, we have Add the following, on lines 329-354,and lines 366-385.

In an 8-week trial enrolling 60 patients with primary hyperlipidemia, subjects were randomized into an experimental and a control group, with the experimental group re-ceiving a daily preparation containing NK 2,000 FU and the control group given a placebo. After a 56-day intervention, there were significant mean absolute changes in total choles-terol (TC), low-density cholesterol (LDL-C), non-high-density cholesterol (non-HDL-C), and low-density cholesterol to high-density cholesterol ratio (LDL-C to HDL-C ratio), with values of −0.52 (95% CI: −0.51 to −0.54) mmol/L, −0.43 (95% CI: −0.45 to −0.41) mmol/L, −0.52 (95% CI: −0.52 to −0.52) mmol/L, and −0.29 (95% CI: −0.30 to −0.28) mmol/L, respec-tively, between the two groups. The results showed that serum total cholesterol (TC) levels in the experimental group were reduced by an average of about 10%, triglycerides (TG) were reduced by about 12%, low-density lipoprotein cholesterol (LDL-C) was reduced by about 11%, and high-density lipoprotein cholesterol (HDL-C) was elevated by about 8%, P <0.05 and the difference was statistically significant; in the control group, there was no significant change in the lipid indices[20].

In another study, 30 patients with carotid intima-media thickening underwent a 3-month intervention in which the experimental group was given NK (1500 FU/day) in combination with conventional lipid-lowering drugs, and the control group was given only conventional lipid-lowering drugs. At the end of the experiment, the reduction in ca-rotid intima-media thickness (IMT) was more significant in the experimental group than in the control group, there was a significant reduction in CCAIMT and carotid plaque size in both groups compared with the baseline before treatment. The carotid plaque size and CCA-IMT reduced from(0.25 ± 0.12) cm2 to (0.16 ± 0.10) cm2 and from (1.13 ± 0.12)mm to (1.01 ± 0.11) mm, repectively. The reduction in the NK group was significantly profound (P <0.01, 36.6% reduction in plaque size in NK group versus 11.5% change in stat

The results of the study are impressive, atherosclerotic plaque patients with 21.7% reduction in the thickness of the inner intima-media, with an effective rate of 77.7%, and 36.0% reduction in the area of plaque, with an effective rate of 66.5%, indicating that NK can effectively inhibit the progression of atherosclerosis. After 12 months of daily NK consumption at a dose of 10,800 FU, a significant reduction in TG, TC, and LDL-C (P < 0.01) was evident compared to the values before treatment. Furthermore, NK also had the effect of increasing HDL-C (15.8% increase, P < 0.01). The levels of TC, TG, LDL-C, and HDL-C improved in 95.4, 85.2, 84.3, and 89.1% of the participants, respectively, after 12 months of NK use (Table 3). NK administration for 12 months led to a decrease of 15.9, 15.3, and 18.1% in TC, TG, and LDL-C, respectively. In terms of lipid regulation, total cholester-ol in hyperlipidemic patients was reduced by 15.9%, with an effective rate of 95.4%; tri-glycerides were reduced by 15.7%, with an effective rate of 85.2%; LDL cholesterol was re-duced by 18.1%, with an effective rate of 84.3%; and HDL cholesterol was increased by 15.8%, with an effective rate of 89.1%, which comprehensively improved the lipid profile and no obvious adverse reactions related to the use of NK were found in the whole study process. No obvious adverse reactions related to the use of NK were found in the whole study, and there was no significant difference in the efficacy between male and female subjects, which fully confirmed the effectiveness and safety of this dose of NK in the pre-vention and treatment of cardiovascular diseases, and provided key evidence for the pro-motion of clinical application.

Comments 5: [ Additionally, the limitations of the studies are not adequately discussed. Reorganizing the manuscript for logical flow and providing a more critical evaluation of the data will strengthen its scientific value. I recommend major revision before further consideration.]

Response 5: Thank you for pointing this out. We agree with this comment. Therefore, we have Add the following, on lines 645-691.

9.Prospects for Future Research Directions

The limitations of the research on the lipid-lowering effect of nattokinase are mainly reflected in the research design, mechanism of action, sample size and clinical transformation. From the perspective of research design, most of the current studies are based on in vitro experiments or animal models. The number of human clinical studies, especially large-sample, long-term follow-up randomized controlled trials (RCTs), is limited, resulting in doubts about the reliability and universality of the conclusions. For example, the model induced by high-fat diet commonly used in animal experiments is different from the pathophysiological mechanism of human hyperlipidemia, and the dose, dosage form (such as whether it is enteric-coated) and administration route of nattokinase may affect the experimental results. However, the relevant comparative studies are insufficient, and it is difficult to determine the optimal intervention plan.

9.1. Exploration of mechanism of action

The specific targets and signaling pathways of nattokinase in lowering blood lipid have not been fully elucidated. Most of the existing studies focus on its thrombolytic and anticoagulant effects, and there is a lack of systematic research on its direct effects on lipid metabolism (such as whether to regulate liver cholesterol synthesis, promote fatty acid β-oxidation or affect intestinal lipid absorption). Some studies have speculated that it may indirectly affect lipid metabolism by improving vascular endothelial function or anti-inflammatory effects, but the causal relationship has not been verified by mechanism experiments, resulting in a lack of solid theoretical basis.

9.2. Sample size and heterogeneity

Published human studies are mostly small samples (such as dozens of subjects), and the inclusion criteria are not uniform. There is a lack of stratified analysis of age, gender, underlying diseases (such as diabetes, hypertension) and medication history (such as statins), and it is difficult to exclude confounding factors. For example, some studies did not control lifestyle factors such as diet and exercise, resulting in the inability to accurately assess the independent role of nattokinase. In addition, the sources (such as fermentation products of different strains), purity and dose of nattokinase in different studies vary greatly, and the results are not comparable, so it is difficult to form a consensus through meta-analysis.

9.3. Clinical transformation level

As a natural product, the quality control standards (such as enzyme activity units and impurity residues) of nattokinase have not been unified, and the effectiveness and safety of commercially available products are uneven, which may affect the repeatability of the research results. At the same time, the potential risks of long-term use of nattokinase (such as bleeding tendency and liver burden) lack data support, especially the lack of research on the interaction with existing lipid-lowering drugs, which limits its clinical application.

9.4. Clinical transformation level

As a natural product, the quality control standards (such as enzyme activity units and impurity residues) of nattokinase have not been unified, and the effectiveness and safety of commercially available products are uneven, which may affect the repeatability of the research results. At the same time, the potential risks of long-term use of nattokinase (such as bleeding tendency and liver burden) lack data support, especially the lack of research on the interaction with existing lipid-lowering drugs, which limits its clinical application.

Reviewer 3 Report

Comments and Suggestions for Authors

This paper is important because it provides a comprehensive synthesis of current evidence on the lipid-lowering mechanisms and clinical potential of nattokinase, a natural enzyme with emerging therapeutic relevance for dyslipidemia and cardiovascular disease prevention, offering a promising alternative or adjunct to conventional lipid-lowering therapies.

  1. The manuscript should be revised to eliminate typographical errors. As an example. There are many places where a period, spaces, etc. are missing.
    • As an example.Line 9 “ dyslipidemiaThe prevalence” it should be “   The prevalence”
    • “natto.Discovered in 1980…” it should be  “natto. Discovered in 1980”
  2. In figure Figure 1 caption, “Chemical struc-ture of Nattokinas” it should be “Nattokinase”
  3. The heading 5.1 and 5.2 are the same.

5.1. Review of Early Small-Scale Clinical Trials

5.2. Review of Early Small-Scale Clinical Trials

  1. heading 3.2 “Hemoepidemiologic findings” is not standard in cardiovascular literature, replace with “Epidemiologic findings”
  2. Some sentences are somewhat vague, phrases like “white or light yellow solid” and “without obvious odor” they should be omitted or qualified more precisely to maintain a scientific tone.(lines 56-57)
  3. Please revise the term “celiac micro-particles (CM)” to “chylomicrons (CM),” as “celiac micro-particles” is not a normal term in lipid metabolism, and the correct physiological term for particles that transport triglycerides is “chylomicrons.” (line s 174-175)
  4. The sentence “red currant is rich in natural statins” should be corrected to “red yeast rice is rich in natural statins.” Red currants do not contain statins, while red yeast rice is well-documented as a source of monacolin K, a natural compound with statin-like activity. This is also consistent with the cited reference [31], which studied red yeast rice in combination with nattokinase.
  5. In Section 6.1, the use of the term “Biotics” appears to be a typographical or conceptual error in the context of lipid-lowering therapy. The sentence in question describes agents that primarily target elevated triglyceride levels and low HDL-C by activating peroxisome proliferator-activated receptor alpha (PPAR-α), and also references fenofibrate specifically. These characteristics are consistent with fibrates, a well-established pharmacological class used to manage hypertriglyceridemia and mixed dyslipidemia. “Biotics,” by contrast, is a broad and nonspecific term typically associated with probiotics, prebiotics, or synbiotics, which do not share the described mechanisms of action. we recommend replacing “Biotics” with “Fibrates” in this context.
  6. I recommend that the authors rewrite their research aim using the PICO framework (Population, Intervention, Comparison, Outcome). eg, the research aim might be reframed as: “To evaluate the lipid-lowering efficacy and safety of nattokinase (Intervention) in individuals with dyslipidemia or cardiovascular risk factors (Population), compared with placebo or conventional lipid-lowering agents (Comparison), focusing on outcomes such as serum lipid profile improvement and cardiovascular risk reduction (Outcome).”

  1. In the description of NK’s molecular characteristics link between structural motifs and enzymatic activity could be slightly expanded. Eg how calcium-binding sites contribute to stability or activity under physiological conditions.
  2. The physicochemical profile include pH and temperature stability, isoelectric point, and tolerance to freeze-thaw cycles. Authors should  explain  why these features matter for formulation or shelf-life (eg  in capsule or beverage forms)
  3. The authors also describe how metal ions affect NK activity, correctly noting that ions such as Hg²⁺ completely inactivate the enzyme while others (e.g., Ca²⁺) enhance it. These findings are relevant but could be strengthened by briefly indicating the mechanism of action (e.g., interaction with active site residues or structural destabilization).
  4. In Subsection 2.2, the description of NK production from Bacillus subtilis var. natto is god. Briefly discusse the trade-offs between native fermentation (eg for food use) and recombinant production (egfor pharmaceuticals), including implications for activity, purity, and regulatory compliance.

BIBLIOGRAHY The bibliography need a deep and strong revisions

  1. The reference list in its current form contains numerous entries that do not conform to the formatting guidelines required by Nutrients, which follow the ACS style as outlined in the journal’s author instructions https://www.mdpi.com/journal/nutrients/instructions). To meet these requirements, all references should include full article or source titles, properly abbreviated journal names, consistent author formatting (surname followed by initials), and complete bibliographic details, including year, volume, and page range or article number when applicable.
  2. Several references are missing critical information. These include numerous Chinese academic theses (e.g., references 5, 6, 7, 8, 21, 23, and 24), which currently lack degree level, institution name, city, and date of completion, as required for thesis citations. Also include URL if available.
  3. reference 1 cites a news source without an author or publication date; this should either be removed or properly formatted as a web reference, including the full URL and access date.
  4. Journal articles such as references 3, 12, 13, 15, 18, and 31 are missing full titles and sometimes journal abbreviations or page/article numbers. All journal citations should follow ACS format: Author(s); Title of the article; Journal Name (abbreviated); Year; Volume; Page range or article ID. Please ensure that all articles include full, correct titles and adhere to this structure.
  5. Some entries (eg, reference 20, which cites a clinical trial) refer to databases without providing a full citation or persistent identifier. These should be cited as web resources, including a full title, working URL, and access date, as required by the journal
  6. The author names in the reference list are presented inconsistently. Some entries use the full name in the Chinese naming order (eg Ren Nina), while others use the correct ACS style (surname followed by initials, eg, Chen H.). Please revise the references (5, 6, 7, 8, 9, 10, 11, 12, 13, 21, 43, 44, 45, 46, 47 )  to match the required ACS format as described in the Nutrients Author Instructions. Specifically, all author names should be presented as “Surname, Initials

  1. The manuscript should be revised to eliminate typographical errors. As an example. There are many places where a period, spaces, etc. are missing.
    • As an example.Line 9 “ dyslipidemiaThe prevalence” it should be “   The prevalence”
    • “natto.Discovered in 1980…” it should be  “natto. Discovered in 1980”

  1. In figure Figure 1 caption, “Chemical struc-ture of Nattokinas” it should be “Nattokinase”

  1. The heading 5.1 and 5.2 are the same.

5.1. Review of Early Small-Scale Clinical Trials

5.2. Review of Early Small-Scale Clinical Trials

  1. 2 heading “Hemoepidemiologic findings” is not standard in cardiovascular literature, replace with “Epidemiologic findings”
  2. Some sentences are somewhat vague, phrases like “white or light yellow solid” and “without obvious odor” they should be omitted or qualified more precisely to maintain a scientific tone.(lines 56-57)

  1. Please revise the term “celiac micro-particles (CM)” to “chylomicrons (CM),” as “celiac micro-particles” is not a normal term in lipid metabolism, and the correct physiological term for particles that transport triglycerides is “chylomicrons.” (line s 174-175)
  2. The sentence “red currant is rich in natural statins” should be corrected to “red yeast rice is rich in natural statins.” Red currants do not contain statins, while red yeast rice is well-documented as a source of monacolin K, a natural compound with statin-like activity. This is also consistent with the cited reference [31], which studied red yeast rice in combination with nattokinase.
  3. In Section 6.1, the use of the term “Biotics” appears to be a typographical or conceptual error in the context of lipid-lowering therapy. The sentence in question describes agents that primarily target elevated triglyceride levels and low HDL-C by activating peroxisome proliferator-activated receptor alpha (PPAR-α), and also references fenofibrate specifically. These characteristics are consistent with fibrates, a well-established pharmacological class used to manage hypertriglyceridemia and mixed dyslipidemia. “Biotics,” by contrast, is a broad and nonspecific term typically associated with probiotics, prebiotics, or synbiotics, which do not share the described mechanisms of action. we recommend replacing “Biotics” with “Fibrates” in this context.

  1. I recommend that the authors rewrite their research aim using the PICO framework (Population, Intervention, Comparison, Outcome). eg, the research aim might be reframed as: “To evaluate the lipid-lowering efficacy and safety of nattokinase (Intervention) in individuals with dyslipidemia or cardiovascular risk factors (Population), compared with placebo or conventional lipid-lowering agents (Comparison), focusing on outcomes such as serum lipid profile improvement and cardiovascular risk reduction (Outcome).”

  1. In the description of NK’s molecular characteristics link between structural motifs and enzymatic activity could be slightly expanded. Eg how calcium-binding sites contribute to stability or activity under physiological conditions.
  2. The physicochemical profile include pH and temperature stability, isoelectric point, and tolerance to freeze-thaw cycles. Authors should  explain  why these features matter for formulation or shelf-life (eg  in capsule or beverage forms)

  1. The authors also describe how metal ions affect NK activity, correctly noting that ions such as Hg²⁺ completely inactivate the enzyme while others (e.g., Ca²⁺) enhance it. These findings are relevant but could be strengthened by briefly indicating the mechanism of action (e.g., interaction with active site residues or structural destabilization).

  1. In Subsection 2.2, the description of NK production from Bacillus subtilis var. natto is god. Briefly discusse the trade-offs between native fermentation (eg for food use) and recombinant production (egfor pharmaceuticals), including implications for activity, purity, and regulatory compliance.

BIBLIOGRAPH

  1. The reference list in its current form contains numerous entries that do not conform to the formatting guidelines required by Nutrients, which follow the ACS style as outlined in the journal’s author instructions https://www.mdpi.com/journal/nutrients/instructions). To meet these requirements, all references should include full article or source titles, properly abbreviated journal names, consistent author formatting (surname followed by initials), and complete bibliographic details, including year, volume, and page range or article number when applicable.
  2. Several references are missing critical information. These include numerous Chinese academic theses (e.g., references 5, 6, 7, 8, 21, 23, and 24), which currently lack degree level, institution name, city, and date of completion, as required for thesis citations. Also include URL if available.

  1. reference 1 cites a news source without an author or publication date; this should either be removed or properly formatted as a web reference, including the full URL and access date.
  2. Journal articles such as references 3, 12, 13, 15, 18, and 31 are missing full titles and sometimes journal abbreviations or page/article numbers. All journal citations should follow ACS format: Author(s); Title of the article; Journal Name (abbreviated); Year; Volume; Page range or article ID. Please ensure that all articles include full, correct titles and adhere to this structure.
  3. Some entries (eg, reference 20, which cites a clinical trial) refer to databases without providing a full citation or persistent identifier. These should be cited as web resources, including a full title, working URL, and access date, as required by the journal
  4. The author names in the reference list are presented inconsistently. Some entries use the full name in the Chinese naming order (eg Ren Nina), while others use the correct ACS style (surname followed by initials, eg, Chen H.). Please revise the references (5, 6, 7, 8, 9, 10, 11, 12, 13, 21, 43, 44, 45, 46, 47 )  to match the required ACS format as described in the Nutrients Author Instructions. Specifically, all author names should be presented as “Surname, Initials

Author Response

Response 1:Thank you for pointing this out. We agree with this comment. Therefore, we have made the following change.[Natto is a traditional Japanese fermented food, which is made from soybeans fermented by Bacillus natto. During the fermentation process, natto will produce a variety of biologically active substances, including nattokinase.]

we have add the following in lines 10-12 “[updated text in the manuscript if necessary]”

Comments 2: In figure Figure 1 caption, “Chemical struc-ture of Nattokinas” it should be “Nattokinase”

Response 2: Thank you for pointing this out. We agree with this comment. Therefore, we have made the following change.Figure 1. The structure of NK is shown in Fig. A: Spatial structure of Nattokinase. B: Chemical structure of Nattokinase.

we have add the following in lines 171-172 “[updated text in the manuscript if necessary]”

Comments 3: In The heading 5.1 and 5.2 are the same.

5.1. Review of Early Small-Scale Clinical Trials

5.2. Review of Early Small-Scale Clinical Trials

Response 3: Thank you for pointing this out. We agree with this comment. Therefore, we have made the following change.5.1. Review of Early Small-Scale Clinical Trials

5.2. Review of Early large-Scale Clinical Trials.

we have add the following in lines 323 and 355“[updated text in the manuscript if necessary]”

Comments 4: heading 3.2 “Hemoepidemiologic findings” is not standard in cardiovascular literature, replace with “Epidemiologic findings”

Response 4: Thank you for pointing this out. We agree with this comment. Therefore, we have made the following change.Epidemiologic findings  

we have add the following in lines 256“[updated text in the manuscript if necessary]”

Comments 5: Some sentences are somewhat vague, phrases like “white or light yellow solid” and “without obvious odor” they should be omitted or qualified more precisely to maintain a scientific tone.(lines 56-57)

Response 5: Thank you for pointing this out. We agree with this comment. Therefore, we have made the following change.In terms of physicochemical properties, NK is in the form of light yellow solid, with good water solubility.

we have add the following in lines 142-143“[updated text in the manuscript if necessary]”

Comments 6: Please revise the term “celiac micro-particles (CM)” to “chylomicrons (CM),” as “celiac micro-particles” is not a normal term in lipid metabolism, and the correct physiological term for particles that transport triglycerides is “chylomicrons.” (line s 174-175)

Response 6: Thank you for pointing this out. We agree with this comment. Therefore, we have made the following change.[ For lipoprotein metabolism, NK enhances the activity of lipoprotein lipase (LPL), a key enzyme in triglyceride hydrolysis, which breaks down triglycerides in chylomicrons (CM) and very low-density lipoproteins (VLDL) into fatty acids for tissue uptake and utilization[19], and reduces plasma triglyceride levels. ]

we have add the following in lines 312“[updated text in the manuscript if necessary]”

Comments 7: The sentence “red currant is rich in natural statins” should be corrected to “red yeast rice is rich in natural statins.” Red currants do not contain statins, while red yeast rice is well documented as a source of monacolin K, a natural compound with statin-like activity. This is also consistent with the cited reference [31], which studied red yeast rice in combination with nattokinase.

Response 7: Thank you for pointing this out. We agree with this comment. Therefore, we have made the following change.[ For natural product combination applications, red yeast rice is rich in natural statins, which complement the NK combination.]

we have add the following in lines 467“[updated text in the manuscript if necessary]”

Comments 8: In Section 6.1, the use of the term “Biotics” appears to be a typographical or conceptual error in the context of lipid-lowering therapy. The sentence in question describes agents that primarily target elevated triglyceride levels and low HDL-C by activating peroxisome proliferator-activated receptor alpha (PPAR-α), and also references fenofibrate specifically. These characteristics are consistent with fibrates, a well-established pharmacological class used to manage hypertriglyceridemia and mixed dyslipidemia. “Biotics,” by contrast, is a broad and nonspecific term typically associated with probiotics, prebiotics, or synbiotics, which do not share the described mechanisms of action. we recommend replacing “Biotics” with “Fibrates” in this context.

Response 8: Thank you for pointing this out. We agree with this comment. Therefore, we have made the following change.Fibrates mainly target elevated triglycerides and low HDL-C anemia by activating peroxisome proliferator activated receptor alpha (PPAR-α), accelerating VLDL catabolism, lowering triglyceride levels, and elevating HDL-C [29]. Like fenofibrate, it is indicated for hypertriglyceridemic patients with triglycerides > 5.65 mmol/L to prevent acute pancreatitis attacks. However, fibrates also carry potential risks of gastrointestinal distress, hepatic and renal impairment, rhabdomyolysis, and limited LDL-C reduction when used alone[30].We have already emphasized in the manuscript.

we have add the following in lines 435-441“[updated text in the manuscript if necessary]”

Comments 9: I recommend that the authors rewrite their research aim using the PICO framework (Population, Intervention, Comparison, Outcome). eg, the research aim might be reframed as: “To evaluate the lipid-lowering efficacy and safety of nattokinase (Intervention) in individuals with dyslipidemia or cardiovascular risk factors (Population), compared with placebo or conventional lipid-lowering agents (Comparison), focusing on outcomes such as serum lipid profile improvement and cardiovascular risk reduction (Outcome).”

Response 9: Thank you for pointing this out. We agree with this comment. Therefore, we have made the following change.The main component of natto lipid-lowering is nattokinase. Although these components have a certain role in dissolving thrombus and reducing blood viscosity, which may have certain benefits for blood lipid regulation, the mechanism of action is relatively simple. The mechanism of dyslipidemia is complex and involves many factors. These components in natto alone are difficult to comprehensively and effectively deal with all the links leading to dyslipidemia. The content of effective components is limited : the content of lipid-lowering active components in natto is relatively low. In order to achieve a significant lipid-lowering effect, it may be necessary to eat a large amount of natto. However, eating a lot of natto may bring other problems, such as excessive intake of protein, purine, etc., adversely affect health, especially for people with kidney disease, gout and other basic diseases.We have already emphasized in the manuscript.

we have add the following in lines 142-143“[updated text in the manuscript if necessary]”

Comments 10: In the description of NK’s molecular characteristics link between structural motifs and enzymatic activity could be slightly expanded. Eg how calcium-binding sites contribute to stability or activity under physiological conditions.

Response 10: Thank you for pointing this out. We agree with this comment. Therefore, we have made the following change.On the level ofmolecular biology, DNA family shuffling is one of the ways to improve the fibrinolytic activity of nattokinase[5].Site-directed mutagenesis is another means to enhanceactivity and oxidative stability of NK[6,7]. During the NK purification process, the traditional protein expression and purification technology were changed and improved for better activity, but the results were less useful[8]. Beyond that, chemical modification has become an effective and popular approach to improve nattokinase activity and stability. NK was first well immobilized onto magnetic nanoparticles Fe3O4 in the presence of 1-[3-(dimethylamino)propyl]-3-ethylcarbodiimide (EDC), and it turned out that it had much higher thrombolytic activity, even higher than the pure NK[9]. Immobilization of nattokinase upon polyhydroxybutyrate (PHB) nanoparticles resulted in a 20% increase in the enzyme activity[10]. The adsorption capacity of nanosilver (AgNPs) on nattokinase enhanced heat stability and anticoagulant effect of NK[11]. The effect of various metal ions, such as Ag(I),K(I), Na(I), Ba(II), Ca(II), Cd(II), Co(II), Cu(II), Hg(II),Mg(II), Mn(II), Ni(II), Zn(II), Fe(II), Fe(III) and Al(III),had been tested on enzyme activity with a final concentrations from 1 to 10 mmol/L[12].we have add the following in lines 127-141“[updated text in the manuscript if necessary]”

Comments 11: The physicochemical profile include pH and temperature stability, isoelectric point, and tolerance to freeze-thaw cycles. Authors should  explain  why these features matter for formulation or shelf-life (eg  in capsule or beverage forms)

Response 11: Thank you for pointing this out. We agree with this comment. Therefore, we have made the following change.Understanding the pH and temperature stability, isoelectric point, and freeze-thaw cycle tolerance of nattokinase is helpful to select appropriate ingredients and processing conditions in food formulation design. For example, select a suitable buffer system to maintain the pH value of the food within the stable range of nattokinase, or add a protective agent to improve its heat resistance and freeze-thaw cycle resistance. In the process of processing and storage of food containing nattokinase, high temperature, acidic or alkaline environment and multiple freeze-thaw cycles should be avoided to ensure the stability of nattokinase. For example, the use of low-temperature processing and storage technology, or through genetic engineering and other means to improve the heat resistance of nattokinase and freeze-thaw cycle resistance. In addition, the activity of nattokinase is regularly detected during the production and storage of food containing nattokinase to ensure its stability in food. By understanding its physical and chemical properties, we can better formulate quality control standards and detection methods to ensure the functional properties and safety of food.

we have add the following in lines 155-169“[updated text in the manuscript if necessary]”

Comments 12: The authors also describe how metal ions affect NK activity, correctly noting that ions such as Hg²⁺ completely inactivate the enzyme while others (e.g., Ca²⁺) enhance it. These findings are relevant but could be strengthened by briefly indicating the mechanism of action (e.g., interaction with active site residues or structural destabilization).

Response 12: Thank you for pointing this out. We agree with this comment. Therefore, we have made the following change. The answer to this part is the same as Response 10.

we have add the following in lines 127-141“[updated text in the manuscript if necessary]”

Comments 13: In Subsection 2.2, the description of NK production from Bacillus subtilis var. natto is god. Briefly discusse the trade-offs between native fermentation (eg for food use) and recombinant production (egfor pharmaceuticals), including implications for activity, purity, and regulatory compliance.

Response 13: Thank you for pointing this out. We agree with this comment. Therefore, we have made the following change. NK production by natural fermentation mainly depends on the traditional natto fermentation process. NK is produced by natural fermentation of Bacillus subtilis var.natto under specific conditions. The advantage of this method is that the process is relatively simple and the cost is low, which is suitable for large-scale production in the food field. However, the NK activity produced during natural fermentation is relatively low, and it is usually difficult to meet the needs of high activity in the pharmaceutical field. In addition, a variety of metabolites are produced during the natural fermentation process. The presence of these by-products will reduce the purity of NK, which may affect its application in the pharmaceutical field. From the perspective of regulatory compliance, NK produced by natural fermentation is mainly used in food, and its regulatory requirements are relatively relaxed. However, if it is applied to the pharmaceutical field, it is necessary to meet stricter regulatory requirements such as GMP, which may increase additional costs and complexity. In contrast, recombinant production of NK is the use of genetic engineering technology to transform Bacillus subtilis var.natto to improve the yield and activity of NK. For example, by knocking out certain genes ( such as the SP EPR and SP WAPA gene ), unnecessary metabolites can be reduced, thereby increasing the purity of NK. This recombinant production method can significantly improve the activity and yield of NK, making it more suitable for use in the pharmaceutical field. However, the cost of recombinant NK production is relatively high, and more resources need to be invested in genetic engineering, fermentation process optimization and quality control. From the perspective of regulatory compliance, reorganized NK is mainly used in pharmaceuticals, so it is necessary to strictly comply with regulations such as the Good Manufacturing Practice ( GMP ) to ensure the safety, effectiveness and quality controllability of products. This not only increases the supervision cost in the production process, but also puts forward higher requirements for the quality management system of production enterprises. We have already emphasized in the manuscript.

we have add the following in lines 202-228“[updated text in the manuscript if necessary]”

Comments 14-19: BIBLIOGRAHY The bibliography need a deep and strong revisions

Response 14-19: Thank you for pointing this out. We agree with this comment. Therefore, we have made the following change.we have add the following in lines 705-857“[updated text in the manuscript if necessary]”

Round 2

Reviewer 1 Report

Comments and Suggestions for Authors

The authors revised their manuscript according to the referee's comments.

Reviewer 2 Report

Comments and Suggestions for Authors

Well done!

Reviewer 3 Report

Comments and Suggestions for Authors

Dear authors thanks for incorporating my points into the manuscript. In my opinion it can be published as it is, in its present form.